**Data Availability Statement:** The data collected from this study cannot be shared publicly because it is co-owned by the Ministry of Health Turkey and WHO. The WHO ethics committee required that data collected from the study be stored in WHO

# Factors affecting patient satisfaction in refugee health centers in Turkey

Monica Zikusooka[1][☯]*, Radysh Hanna[1][☯], Altin Malaj[1][‡], Meliksah Ertem[2][‡], Omur Cinar Elci[1][¤][‡]

1 Refugee Health Programme, WHO Country Office in Turkey, WHO Regional Office for Europe, Turkey,
2 SIHHAT Project, Ministry of Health Turkey, Turkey

☯ These authors contributed equally to this work.
¤ Current address: West Atlantic University School of Medicine, Freeport, Grand Bahama, The Bahamas
‡ AM, ME, and OCE also contributed equally to this work.
* mona@zikusooka.com, mzikusooka@who.int

## Abstract

### Background

Turkey hosts an estimated 3.7 million Syrian refugees. Syrian refugees have access to free primary care provided through Refugee Health Centers(RHC). We aimed to determine factors that influence patient satisfaction in refugee health centers.

### Methods

The study was a cross-sectional quantitative study. A patient survey was administered among 4548 patients attending services in selected 16 provinces in Turkey. A quantitative questionnaire was used to collect information on patient satisfaction and experience in the healthcare facility. Information on "overall satisfaction with health services" was collected on a 5-point Likert scale and dichotomized for analysis. Logistic regression was conducted to identify factors that influenced patient satisfaction.

### Results

We found that 78.2% of the participants were satisfied with the health services they received. Factors related to service quality and communication were significant determinants of patient satisfaction. The strongest predictors of satisfaction were having a sufficient consultation time (AOR: 2.37; 95% CI: 1.76–3.21; p< 0.0001), receiving a comprehensive examination (AOR: 2.01; 95% CI: 1.49–2.70; $p$ < 0.0001) and being treated with respect by the nurse (AOR: 2.08; 95% CI: 1.52–2.85; p< 0.0001).

### Conclusion

Providing integrated, culturally and linguistically sensitive health services is important in refugee settings. The quality of service and communication with patients influence patient satisfaction in refugee health centers. As such, improvements in aspects such as consultation time and the quality of physician-patient interaction are recommended for patient satisfaction.

Turkey country office and be only used for the purpose of this study. A request to access this data can be sent to the WHO representative in Turkey at "eurowhotur@who.int" (https://www.euro.who.int/en/countries/turkey/contact-us).

**Funding:** The research was conducted under a health services project funded by the EU through the EU Regional Trust Fund in Response to the Syrian Crisis that ended in June 2021. World Health Organization Turkey office was the award recepient;none of the authors received a specific award. The funders had no role in study design, data collection and analysis, decision to publish, or preparation of the manuscript.

**Competing interests:** The authors have declared that no competing interests exist.

# Introduction

Turkey currently hosts almost 3.7 million Syrians, of whom 46% are women, and 14% are children aged 0–4 years [1]. Globally, refugees and migrants often face challenges in accessing health care, including language and cultural differences [2–4], low health literacy [2], difficulties in understanding the health system [3], legal status, lack of awareness of their health rights [4] and financial limitations [2]. In line with international commitments to refugee protection, Turkey has taken steps to ensure access to health services for its large refugee population.

Primary health care is the basis to achieving Universal Health Care [UHC] and the Sustainable Development Goads [5]. In Turkey, primary health care is provided through community health centers and family health centers. As part of the community health centre network, the government established the RHC mechanism with RHCs to provide cultural and language-sensitive primary healthcare services to the Syrian population. Under the Ministry of Health, RHCs are called migrant health centers. Most healthcare providers in RHCs are Syrian nationals. The mechanism includes standard RHCs, extended RHCs and RHTCs. RHCs comprise several refugee health units, with each consisting of a physician and nurse team. Extended RHCs provide additional specialty services, including internal medicine, pediatric, obstetrics and gynecology, oral and dental health, psychosocial support, and simple imaging and laboratory services. There are also seven RHTCs that provide all of the services of extended RHCs and have training facilities for health workers.

Definitions and concepts of patient satisfaction vary. However, examining patients' views on health care and which attributes they value most can provide insights to improve the quality of care and inform strategic decision-making [6, 7]. Satisfied patients are more likely to adhere to treatment plans, which increases the chance of good health outcomes and fewer diagnostic tests and referrals, increasing care efficiency [6, 8]. Satisfied patients are also likely to return or recommend the services they have received to others, thereby helping to improve service utilization [9].

Studies on people-centered care and patient satisfaction have produced a wide body of evidence and analytical tools [10–14]. For refugees, migrants, and asylum seekers, high levels of patient satisfaction were found when health services were provided in specialized units or delivered with language and cultural sensitivity [10–12]. Evidence shows that multiple factors related to the health worker influence patient satisfaction, including technical expertise, interpersonal care (e.g. communication), physical environment, access (i.e. accessibility, availability and cost), organizational characteristics, continuity of care, treatment outcome, and length of consultation with the doctor [6, 14]. In addition, patient characteristics such as age, gender, education, socioeconomic status, marital status, race, religion, geographical characteristics, frequency of visits, length of stay in host country, health status, personality and expectations were also found to influence patient satisfaction, but with inconsistent strength and direction of effect [14]. Although patient satisfaction is a common outcome measure in health care assessments, it may be influenced by patients' expectations as much as by the quality of the care provided. The match between patient expectations and what care is provided also influences patient satisfaction [15, 16].

Within the humanitarian context, assessing the satisfaction of patients who receive services from RHCs in Turkey is critical for accountability to the people most affected by the Syrian conflict. Accountability demands that actions to help people in need are driven by the needs, desires, and capacities of the people affected and implemented respectfully. In this regard, the humanitarian sector has committed to allowing affected populations to provide feedback on the goods and services they have received through humanitarian actions [17, 18].

Some household surveys on the health needs of Syrian refugees in Turkey have assessed utilization and satisfaction with healthcare services [19, 20]. Another study evaluated patient satisfaction with mental health and psychosocial support services in RHTCs [21]. However, to our knowledge, no study had assessed factors that influence patient satisfaction in RHCs across the RHC mechanism. This study aimed to determine patient satisfaction and factors influencing satisfaction among patients who received healthcare services from RHCs.

## Methodology

### Study period and population

The patient survey was conducted between December 2019 and January 2020. Sixteen provinces in Turkey with the highest number of patient consultations in RHCs were selected to achieve a high representation of patients receiving health care services from RHCs. To be included in the study, participants had to be adult patients (aged > 18 years) or an immediate adult caregiver of a patient (child, spouse, or elderly) who received healthcare services in RHCs. Participants also should have had at least one contact with healthcare practitioners that included physical examination, diagnostic test or therapeutic intervention on the day of the interview or within 30 days prior to the interview. Patients under 18 years were excluded if they did not have an adult caregiver or guardian.

### Study design, sample size determination and sampling techniques

The study was a cross-sectional quantitative study. A proportional stratified sampling approach was followed to estimate the required sample size based on the total patient consultations in each province(strata) from 2017 until March 2019. A minimum sample size of 4460 individuals, was calculated using WinPepi (version 11.65) with a 95% CI, 0.05 error margin, and 20% loss to follow-up. The sample size was then distributed proportionally to the volume of consultations in each of the 16 provinces and type of RHCs (Table 1). The RHCs where data was collected were randomly selected, from a list of RHCs provided by the Ministry of Health.

### Data collection

Data was collected using a quantitative questionnaire developed by WHO in the Yemen emergency response; adopted for its suitability to the context of the humanitarian health response. First, the questionnaire was adapted to the study's objectives, then it was adapted to the Syrian Arabic dialect and piloted in RHCs attended by Syrian refugees. The questionnaire collected information on patient characteristics, use of the health facility, patient experience and satisfaction with services. Data was collected through face-to-face interviews in Arabic by trained data assistants. In each RHC, participants were systematically recruited in the reception areas on regular working days, at an interval calculated from the average daily patient load of the facility.

### Ethics statement

The patient survey and its procedures, including the participant consent process, were reviewed and approved by the WHO Ethical Review Committee, Gazi University Ethical Board and Ministry of Health Ethical Board in Turkey. The consent form was read in Arabic to all participants that met the inclusion criteria. Responses were recorded before administering the interview to only those who to agreed to participate in the survey. Verbal instead of written consent was sought because of the high illiteracy level in the study population.

**Table 1. Sample distribution by province and type of RHC.**

| Province | Sample estimation | | | | | |
|---|---|---|---|---|---|---|
| | Number of Refugee health units | | | Number of Patients | | |
| | RHCs | E-RHCs | RHTCs | RHCs | E-RHCs | RHTCs |
| Adana | 5 | 7 | 0 | 150 | 210 | 0 |
| Ankara | 1 | 2 | 1 | 65 | 130 | 65 |
| Bursa | 4 | 3 | 0 | 120 | 90 | 0 |
| Gaziantep | 3 | 2 | 1 | 150 | 100 | 50 |
| Hatay | 6 | 4 | 1 | 450 | 300 | 75 |
| Istanbul | 7 | 8 | 1 | 210 | 240 | 30 |
| Izmir | 2 | 1 | 1 | 116 | 58 | 58 |
| Kahramanmaraş | 4 | 5 | 0 | 120 | 150 | 0 |
| Kayseri | 2 | 2 | 0 | 60 | 60 | 0 |
| Kilis | 1 | 9 | 0 | 30 | 270 | 0 |
| Konya | 2 | 3 | 0 | 60 | 90 | 0 |
| Malatya | 1 | 1 | 0 | 30 | 30 | 0 |
| Mardin | 1 | 0 | 0 | 37 | 0 | 0 |
| Mersin | 3 | 2 | 1 | 93 | 62 | 31 |
| Osmaniye | 1 | 4 | 0 | 30 | 120 | 0 |
| Şanlıurfa | 4 | 4 | 1 | 232 | 230 | 58 |
| Total | 47 | 57 | 7 | 1953 | 2140 | 367 |
| Total Sample | | | | 4460 | | |

E-RHC: extended RHC.

## Study variables

Based on a literature review and the context of Syrian refugees in Turkey, study variables were identified and categorized into four clusters: 1. patient characteristics: age, gender, education and year of arrival in Turkey; 2. accessibility of healthcare services: commuter time to reach the RHC; 3. communication: healthcare provider explains medical tests, doctor's explanation of medical condition, healthcare provider's explanation of the danger signs; and 4. quality of service: healthcare provider's time spent with the patient, healthcare provider's administration of a comprehensive examination, healthcare provider's attitude towards the patient, waiting time and type of RHC. Information on variables in the communication and service clusters was collected on a five-point Likert scale (strongly disagree, disagree, neither agree nor disagree, agree, strongly agree) and re-categorized into two for statistical analyses: the first three responses (strongly disagree, disagree, neither agree nor disagree) were categorized as "disagree," and the last two (agree, strongly agree) as "agree". Similarly, patient responses for the statement "Overall, the healthcare services I have been receiving are satisfactory" were collected on a five-point Likert scale and re-categorized as two: "disagree"—unsatisfied and "agree"–satisfied for analysis. Collapsing a 5-point scale into a dichotomous or trichotomous scale during data analysis has been found to work well [22].

## Data analysis

Descriptive analyses were conducted to describe the distribution of sociodemographic characteristics and other study variables. Patient experiences and satisfaction were analysed both overall and for the different facility types. Logistic regressions were conducted to identify factors that influenced patient satisfaction. To fit the logistic regression models, variables with a

significant influence on patient satisfaction ($p < 0.05$) were included, and AORs were calculated with 95% CIs. In the first model fitted, each variable with a significant influence on patient satisfaction was adjusted for patient characteristics (age, gender, education, year of arrival in Turkey), whereas in the second model, all variables that influenced patient satisfaction and patient characteristics were adjusted by including them in the model. Data analysis was performed using IBM SPSS Statistics version 25.0.

## Results

### Sociodemographic characteristics of participants

More than 70% of participants had arrived in Turkey after 2013, and 27.3% had arrived after 2016 (Table 2). The average household size was 5.9 people. Nearly two thirds (64.5%) of respondents were women. Most participants (81.5%) were aged under 45 years. Almost a quarter of the respondents (23.7%) were illiterate (not able to read or write) and nearly half (48.4%) had completed primary education only. Overall, about a quarter of respondents (23.9%) were currently employed, but the proportion was higher for men than for women (52.3% vs 8.2%). Regarding employment sectors, half of employed respondents (50.5%) were working in sales and services, 13.8% in agriculture and 12.4% in teaching. Most male respondents were employed in the sales and services sector (57.6%), and similar proportions of female respondents were working in the teaching (28.7%), sales and services (25.4%), and agricultural (23.8%) sectors.

### Patient satisfaction

When asked about their overall level of satisfaction with the healthcare services that they had received at the RHC, 78.2% of all respondents said that they were satisfied (80.1% of men and

**Table 2. Sociodemographic characteristics of respondents.**

| Characteristic | Number (n) | Percentage (%) |
|---|---|---|
| Gender (n = 4533) | | |
| Men | 1608 | 35.5 |
| Women | 2925 | 64.5 |
| Age, years (n = 4533) | | |
| 18–29 | 1,794 | 39.58 |
| 30–39 | 1,484 | 32.74 |
| 40–49 | 723 | 15.95 |
| 50–59 | 370 | 8.16 |
| 60 and above | 162 | 3.57 |
| Education level (n = 4505) | | |
| No education | 1069 | 23.7 |
| Completed primary education | 2180 | 48.4 |
| Completed secondary education | 762 | 16.9 |
| University degree/equivalent or higher | 494 | 11.0 |
| Employment status (n = 4522) | | |
| Working | 1080 | 23.9 |
| Not working | 3442 | 76.1 |
| Year of migration (n = 4528) | | |
| ≤ 2013 | 1205 | 26.6 |
| 2014 | 1057 | 23.3 |
| 2015 | 1028 | 22.7 |
| ≥ 2016 | 1238 | 27.3 |

**Table 3. Patient satisfaction with the RHC services, by demographic characteristic.**

| Characteristic | Dissatisfied | | Satisfied | | P[b] value |
|---|---|---|---|---|---|
| | n | % | n | % | |
| Gender | | | | | |
| Male | 320 | 19.9 | 1,286 | 80.1 | 0.027 |
| Female | 665 | 22.8 | 2,257 | 77.2 | |
| Age, years | | | | | |
| 18–29 | 423 | 23.6 | 1,369 | 76.4 | <0.0001 |
| 30–39 | 423 | 22.3 | 1,475 | 77.7 | |
| 40–49 | 119 | 17.6 | 557 | 82.4 | |
| 50–59 | | | | | |
| 60 and above | 20 | 12.4 | 142 | 87.7 | |
| Education | | | | | |
| No education | 181 | 16.9 | 888 | 83.1 | <0.0001 |
| Completed primary | 480 | 22.1 | 1,697 | 78.0 | |
| Completed secondary | 178 | 23.4 | 583 | 76.6 | |
| University degree or higher | 134 | 27.2 | 359 | 72.8 | |
| Employment status | | | | | |
| Employed | 251 | 23.3 | 827 | 76.7 | 0.147 |
| Unemployed | 729 | 21.2 | 2,710 | 78.8 | |
| Arrival in Turkey | | | | | |
| < = 2013 | 229 | 19.0 | 976 | 81.0 | 0.003 |
| 2014 | 223 | 21.2 | 831 | 78.8 | |
| 2015 | 262 | 25.5 | 766 | 74.5 | |
| = >2016 | 269 | 21.8 | 967 | 78.2 | |
| Type of facility | | | | | |
| Standard RHCs | 413 | 20.0 | 1,651 | 80.0 | <0.0001 |
| Extended RHCs | 347 | 20.4 | 1,358 | 79.7 | |
| Training RHCs | 225 | 29.6 | 534 | 70.4 | |

[b] Pearson's chi-squared test.

77.2% of women (p<0.05; Table 3). Compared with the other age groups, respondents aged 60 years and over were significantly more satisfied with the healthcare services that they had received at the RHC (p< 0.001). Higher proportions of respondents with no education and those who had arrived in Turkey in or before 2013 were satisfied compared with the other subgroups.

## Factors influencing patient satisfaction and experience

**Patient characteristics.** Both gender and age had a significant effect on patient satisfaction. In binomial logistic regression comparisons, the following groups were more likely to be satisfied with the health services they had received at RHCs: men, older people, people with lower education levels and people who had arrived in Turkey before 2013. However, none of the patient characteristics were found to significantly influence patient satisfaction in the multiple regression analysis.

**Accessibility.** Accessibility was measured as the time taken for patients to reach a health facility. Using this measure, the accessibility of health services was significantly associated with patient satisfaction. Respondents with longer journey times to reach the health facility were

**Table 4. Multiple logistic regression analysis of RHC characteristics that might influence patient satisfaction.**

| Variable | Unadjusted | | | Adjusted[a] | | | Adjusted[b] | | |
|---|---|---|---|---|---|---|---|---|---|
| | OR | 95% CI | p value | AOR | 95% CI | p value | AOR | 95% CI | p value |
| **Accessibility** | | | | | | | | | |
| Time to reach RHC, minutes (Ref: 0–15) | | | | | | | | | |
| 16–30 | 0.71 | 0.60–0.82 | 0.001 | 0.70 | 0.60–0.82 | <0.0001 | 0.80 | 0.62–1.02 | 0.076 |
| 31–45 | 0.49 | 0.36–0.66 | <0.0001 | 0.50 | 0.37–0.68 | <0.0001 | 0.54 | 0.33–0.88 | 0.013 |
| > 45 | 0.66 | 0.47–0.93 | 0.017 | 0.66 | 0.47–0.94 | 0.020 | 0.92 | 0.53–1.61 | 0.777 |
| **Communication** | | | | | | | | | |
| The health worker explained the reason for medical tests (Ref: disagree) | 7.53 | 6.36–8.91 | <0.0001 | 7.39 | 6.24–8.77 | <0.0001 | 1.93 | 1.48–2.53 | <0.0001 |
| The doctor spent time explaining my medical condition (Ref: disagree) | 8.93 | 7.56–10.56 | <0.0001 | 8.88 | 7.50–10.52 | <0.0001 | 1.7 | 1.24–2.31 | 0.001 |
| Medication side-effects were explained (Ref: disagree) | 3.82 | 3.16–4.61 | <0.0001 | 3.81 | 3.15–4.61 | <0.0001 | 1.53 | 1.16–2.02 | 0.002 |
| The health worker told me what danger signs related to the diagnosis to look out for (Ref: disagree) | 4.53 | 3.87–5.31 | <0.0001 | 4.56 | 3.88–5.36 | <0.0001 | 1.49 | 1.13–1.96 | 0.004 |
| **Quality of service** | | | | | | | | | |
| The health worker took enough time to answer all my questions (Ref: disagree) | 11.06 | 9.31–13.13 | <0.0001 | 10.94 | 9.19–13.02 | <0.0001 | 2.37 | 1.76–3.21 | <0.0001 |
| The health worker was careful to check everything when examining me (Ref: disagree) | 9.92 | 8.39–11.72 | <0.0001 | 9.83 | 8.29–11.65 | <0.0001 | 2.01 | 1.49–2.70 | <0.0001 |
| The doctor treated me with respect (Ref: disagree) | 14.15 | 11.30–17.72 | <0.0001 | 13.42 | 10.70–16.85 | <0.0001 | 1.91 | 1.32–2.77 | 0.001 |
| The nurse treated me with respect (Ref: disagree) | 9.13 | 7.59–10.99 | <0.0001 | 8.75 | 7.25–10.57 | <0.0001 | 2.08 | 1.52–2.85 | <0.0001 |
| **Type of RHC (Ref: standard RHC)** | | | | | | | | | |
| Extended RHC | 0.98 | 0.83–1.15 | 0.794 | 0.99 | 0.85–1.17 | 0.943 | 1.22 | 0.94–1.58 | 0.14 |
| RTHC | 0.59 | 0.49–0.72 | <0.0001 | 0.61 | 0.51–0.75 | <0.0001 | 0.95 | 0.69–1.30 | 0.744 |
| **Waiting time, minutes (Ref: < 20)** | | | | | | | | | |
| 21–60 | 0.44 | 0.37–0.51 | <0.0001 | 0.44 | 0.38–0.52 | <0.0001 | 0.66 | 0.51–0.84 | 0.001 |
| 61–90 | 0.27 | 0.18–0.41 | <0.0001 | 0.30 | 0.20–0.45 | <0.0001 | 0.58 | 0.29–1.17 | 0.127 |
| > 90 | 0.34 | 0.26–0.44 | <0.0001 | 0.35 | 0.26–0.45 | <0.0001 | 0.41 | 0.27–0.64 | <0.0001 |

[a] Model 1.Each variable adjusted for patient characteristics: age, gender, education level and year of arrival in Turkey

[b] Model 2. Fully adjusted–all variables that influenced patient satisfaction and patient characteristics included.

less satisfied ($p < 0.05$). However, when patient characteristics were controlled for in logistic regression analysis, accessibility ceased to be a significant factor($p = 0.05$) (Table 4).

**Communication.** Patient experiences in receiving health information were used to assess communication between the health worker and patient. Respondents who felt that medical tests, medical conditions, medication side-effects and danger signs related to their health condition to look out for at home had been explained were more likely to be satisfied than those who did not ($p < 0.0001$). Respondents who had received explanations about

their medical condition from the doctor were 8.9 times more likely to be satisfied than those who had not (OR: 8.93; 95% CI: 7.56–10.56; p< 0.0001). All communication variables remained significant predictors of patient satisfaction when all the other factors were controlled for. Receiving an explanation of the medical condition from the doctor was the strongest predictor of patient satisfaction in this category (AOR: 1.98; 95% CI: 1.48–2.53; $p < 0.0001$) (Table 4).

**Quality of service.** The influence of quality of service on patient satisfaction was assessed using the participants' assessment of the length of time spent with the health worker, adequacy of the examination, and level of perceived respect from doctors and nurses, along with the waiting time to see a healthcare worker and type of RHC.

Respondents who felt that they spent enough time with the healthcare worker, received a comprehensive examination, and thought they were treated with respect by both doctors and nurses were more likely to be satisfied (p< 0.05). The length of waiting time was also a significant predictor of patient satisfaction ($p < 0.0001$). Respondents who received healthcare services from extended RHCs and RHTCs were less likely to be satisfied than those who received services from standard RHCs. However, when patient characteristics and other factors were controlled for, the type of RHC was not a significant predictor of patient satisfaction. Multiple logistic regression in the fully adjusted model showed that all service-related variables except for the type of RHC were significant predictors of patient satisfaction. The strongest predictors of satisfaction were having a sufficient consultation time (AOR: 2.37; 95% CI: 1.76–3.21; $p < 0.0001$), receiving a comprehensive examination (AOR: 2.01; 95% CI: 1.49–2.70; $p < 0.0001$) and being treated with respect by the nurse (AOR: 2.08; 95% CI: 1.52–2.85; $p < 0.0001$).

## Discussion

Patient satisfaction is becoming an important patient-based outcome measure in health services. Efforts to improve patient satisfaction may lead to improved utilization of health services [23] and better outcomes because satisfied patients may better adhere to treatment plans and have better health-seeking behavior [8, 24].

This study found a similarly high level of patient satisfaction among refugees when compared with previous studies that evaluated healthcare services offered by a specialized unit for refugees or services delivered with sensitivity to language and cultural needs. A German study found a satisfaction level of 84% for patients who visited an integrated care facility in a reception center for asylum seekers and refugees [13]. In another example, an Australian study found high levels of satisfaction among Vietnamese refugees accessing specialized mental health services at a specialized unit for refugees [10]. Another Australian study on an integrated healthcare service for asylum seekers and refugees also found a high level of satisfaction with patients placing high value on integrated care, good relationships with staff, and the availability of interpreting services and bicultural workers [11]. Another study on the health needs of Syrian refugees also found that a satisfaction rate of 65% in respondents who had accessed services from an RHC [20], and a follow-up survey in 2020 found that this rate had increased 66.2% [19]. Both studies showed that patients valued language translation services and integrated care, further indicating that migrant-sensitive healthcare provision could meet patient needs and increase patient satisfaction. In RHCs, doctors and nurses are Syrians which eases communication between the healthcare workers and patients. Although this study did not examine the contribution of language and integrated care to patient satisfaction in RHCs, these factors underpin the RHC mechanism in Turkey and may, therefore, explain the observed high level of patient satisfaction.

## Factors that determine patient satisfaction in RHCs

Healthcare quality factors strongly influence patient satisfaction, including technical care, interpersonal care, physical environment, access (accessibility, availability and finances), organizational characteristics, continuity of care, and outcome of care [14]. This study found that consultation time was the strongest predictor of patient satisfaction. Other studies have shown that consultation time is positively associated with patient satisfaction [12, 23, 25, 26].

Physicians must balance their time with patients against other tasks such as completing electronic medical records, requesting diagnostic tests, writing prescriptions, making phone calls, and sending emails. The time needed for these tasks has increased with increasing computerization and complexity in the primary care system. Owing to an aging population and an increasing prevalence of chronic conditions and other complex clinical issues, physicians may have limited time to provide quality care and meet the expectations of all patients while effectively fulfilling other tasks. Time pressures are greater in facilities with high patient loads, such as RHCs. In a WHO field assessment of the employability of Syrian health workers in Turkey [27], physicians said that they had high workloads.

Similarly, in a job satisfaction survey among health workers in RHCs, 83% and 73% of general and specialist physicians, respectively, reported seeing more than 40 patients per day on average–assuming an eight-hour day, this indicates an average consultation time of fewer than 12 minutes [28]. Therefore, high patient loads mean that consultation times could be short. Short consultations may not allow discussion of the full range of the patient's healthcare concerns and the psychosocial determinants of health, resulting in reduced patient understanding, increased dissatisfaction, and poor adherence to treatment plans [29]. One study argued that making primary care consultations longer (more than 30 minutes for the routine care of complex primary care patients) would probably reduce emergency room and hospital utilization, unnecessary referrals, and unnecessary diagnostic testing and improve satisfaction levels in both patients and health workers [29]. A lower patient-to-physician ratio could reduce workloads for healthcare workers and increase consultation times. Consequently, patient outcomes and satisfaction could be improved, especially in RHTCs, where patients reported the lowest satisfaction with consultation time.

Respect and recognition of patient preferences, needs and values is a core aspect of people-centered care. This study found that being treated with respect by both doctors and nurses significantly influenced patient satisfaction. Doctors and nurses who treat patients in RHCs are Syrian nationals who have been equipped with the knowledge and skills to work in the Turkish primary healthcare system through an adaptation training programme jointly implemented by WHO and the Ministry of Health. As such, patients in RHCs are treated by health workers who are fellow Syrian nationals and have experienced a similar life crisis, which could lead to more empathetic and respectful interactions and, in turn, increase patient satisfaction. Consistent with this study, a positive association between respectful treatment and patient satisfaction was reported previously [30]. In particular, nursing care was highlighted as having a stronger impact on care evaluation by patients [9].

Time spent waiting to see a health worker was significantly associated with patient satisfaction: patients who waited for longer were less likely to be satisfied. Other studies have also demonstrated that waiting time is negatively associated with patient satisfaction [23, 26, 31]. The average waiting time was 30 minutes, although more patients in extended RHCs and RHTCs reported longer waiting times. Beyond reducing patient satisfaction, longer waiting times may cause patients to leave without being seen by a doctor, thereby undermining their access to health care [32]. As health facility service arrangement and patient volume may affect

waiting times, improvements in these areas could reduce the average waiting time and improve patient satisfaction.

Physician–patient communication is a central aspect of diagnosis, treatment and patient support. During discussions with physicians, patients can express their health concerns and ask questions; they may also receive explanations about issues such as their medical condition, which medical tests are needed, side-effects of medications and danger signs to look out for. This study found that explanations on these topics were strong predictors of patient satisfaction. These findings are consistent with other studies that found a positive relationship between physician–patient communication and patient satisfaction [6, 25, 33, 34]. Other studies found a positive association between patient satisfaction and receiving information on their medical condition [24, 29, 35]. Good physician communication with patients has also been established to increase patient adherence to treatment, and training physicians to better communicate with patients also increased patients' adherence to treatment [36]. In this study, patients were mostly dissatisfied with receiving information about medicine side-effects and on danger signs to look out for at home. Refugees and migrants may have specific challenges in using medicines safely, including language and communication issues, cultural issues, and limited health literacy [2]. Overall, supporting physicians in RHCs to improve their communication skills could positively influence patient satisfaction and adherence to treatment.

When other factors were controlled for, sociodemographic factors were not significant predictors of patient satisfaction. In contrast, other studies suggest that sociodemographic factors may be moderate or mediate other determinants of patient satisfaction [22].

## Strengths and limitations

This study was the first assessment of patient satisfaction to be conducted across RHC mechanism. It included a large sample of patients receiving services from all the three types of RHCs in 16 provinces that host the highest number of Syrian refugees in Turkey. Considering that cumulative patient consultation data in RHCs showed that 96% of the consultations were from the 16 provinces, the results of this study are generalizable. Conducting face-to-face interviews in RHCs may have created social desirability but this could have been minimal. Respondents were patients, immediate caregivers (of patients aged under 18 years or were unable to respond) or husbands responding on behalf of their wife because of the patriarchal structure of Syrian refugee families. Although this arrangement was not expected to affect the results because both patient and caregiver were present at the interview, it might have had some effect on responses where the patient's and caregiver's views did not match.

## Conclusions

The high level of patient satisfaction revealed in this study points to the importance of integrated, culturally sensitive health services provided in the patients' own language in RHCs. Although most patients were satisfied with services in RHCs, improvements in physician-patient interactions and communication could empower patients to participate in managing their treatment and overall health. Reducing waiting times could also improve patient satisfaction.

## Supporting information

**S1 Questionnaire.**
(XLSX)

## Acknowledgments

We thank Kanuni Keklik and Özlem Kahraman Tunay of the Migration Health Department, Ministry of Health of the Republic of Turkey, and to Mr Inanc Sogut of the SIHHAT project, Ministry of Health of the Republic of Turkey for their support in conducting this study. We also thank Melda Keçik, Çetin Doğan Dikmen, Pelin Cebeci, Elif Göksu, Nurtaç Kavukcu, Kadriye Küçükbalci and Mustafa Bahadir Sucakli of the WHO Country Office in Turkey and Oguzhan Akyildirim, Pinar Sağlik and Alev Yucel of TANDANS Data Science Consulting for their valuable contributions in the implementation of the study.

## Author Contributions

**Conceptualization:** Monica Zikusooka, Radysh Hanna.

**Data curation:** Monica Zikusooka, Radysh Hanna.

**Formal analysis:** Monica Zikusooka, Omur Cinar Elci.

**Funding acquisition:** Altin Malaj.

**Methodology:** Monica Zikusooka, Radysh Hanna, Meliksah Ertem, Omur Cinar Elci.

**Project administration:** Monica Zikusooka, Radysh Hanna, Altin Malaj, Meliksah Ertem.

**Supervision:** Monica Zikusooka, Radysh Hanna.

**Validation:** Monica Zikusooka, Radysh Hanna, Meliksah Ertem.

**Writing – original draft:** Monica Zikusooka.

**Writing – review & editing:** Monica Zikusooka, Radysh Hanna, Altin Malaj, Meliksah Ertem, Omur Cinar Elci.

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
