## [Decision Letter · Decision Letter 0]

8 Apr 2022

PONE-D-21-25728Factors affecting patient satisfaction in Refugee health centers in TurkeyPLOS ONE

Dear Dr. Zikusooka,

Thank you for submitting your manuscript to PLOS ONE. After careful consideration, we feel that it has merit but does not fully meet PLOS ONE’s publication criteria as it currently stands. Therefore, we invite you to submit a revised version of the manuscript that addresses the points raised during the review process.

Please see the comments from both the reviewers and address them in your revised manuscript. 

We look forward to receiving your revised manuscript.

Kind regards,

Alok Ranjan

Academic Editor

PLOS ONE

https://journals.plos.org/plosone/s/file?id=ba62/PLOSOne_formatting_sample_title_authors_affiliations.pdf".

2. Please include in your Methods section (or in Supplementary Information files) the participating hospitals/institutions. We note that you have reported significance probabilities of 0 in places. Since p=0 is not strictly possible, please correct this to a more appropriate limit, eg 'p<0.0001'.

3. Please provide additional details regarding participant consent. In the ethics statement in the Methods and online submission information, please ensure that you have specified  1) whether the ethics committee approved the verbal/oral consent procedure, 2) why written consent could not be obtained, and 3) how verbal/oral consent was recorded. If your study included minors, please state whether you obtained consent from parents or guardians in these cases. If the need for consent was waived by the ethics committee, please include this information.

4. Please include additional information regarding the survey or questionnaire used in the study and ensure that you have provided sufficient details that others could replicate the analyses. For instance, if you developed a questionnaire as part of this study and it is not under a copyright more restrictive than CC-BY, please include a copy, in both the original language and English, as Supporting Information.

Furthermore, please provide additional information regarding the development and validation of the questionnaire.

5. We note that the grant information you provided in the ‘Funding Information’ and ‘Financial Disclosure’ sections do not match. When you resubmit, please ensure that you provide the correct grant numbers for the awards you received for your study in the ‘Funding Information’ section.

6. We noted in your submission details that a portion of your manuscript may have been presented or published elsewhere. [Some of the resulst of this study are published in a WHO report on Patient satisfaction and experience at migrant health centres in Turkey. This article is a more concise publication of the factors that influence patient satisfaction.] Please clarify whether this [conference proceeding or publication] was peer-reviewed and formally published. If this work was previously peer-reviewed and published, in the cover letter please provide the reason that this work does not constitute dual publication and should be included in the current manuscript.

7. We note that you have indicated that data from this study are available upon request. PLOS only allows data to be available upon request if there are legal or ethical restrictions on sharing data publicly. For more information on unacceptable data access restrictions, please see http://journals.plos.org/plosone/s/data-availability#loc-unacceptable-data-access-restrictions.

Reviewers' comments:

Reviewer's Responses to Questions

**Comments to the Author**

1. Is the manuscript technically sound, and do the data support the conclusions?

Reviewer #1: Partly

Reviewer #2: Yes

2. Has the statistical analysis been performed appropriately and rigorously? 

Reviewer #1: Yes

Reviewer #2: Yes

3. Have the authors made all data underlying the findings in their manuscript fully available?

Reviewer #1: Yes

Reviewer #2: Yes

4. Is the manuscript presented in an intelligible fashion and written in standard English?

Reviewer #1: Yes

Reviewer #2: Yes

5. Review Comments to the Author

Reviewer #1: Healthcare of refugees is a major humanitarian concern and this type of studies should be encouraged to understand the delivery of healthcare services to refugees. Findings of the study resonates the same theoretical principles those required for the batter patient satisfaction.

there is a few concerns which needs further details -

1. Though sample size to measure the patient satisfaction is sufficient but still need more details on the sample size calculation as how they come to sample size of 4548, do they have calculated the sample size for each province separately. How they choose patients visiting health facilities (sampling technique), do they use design effect?

2. Study mostly captures the satisfaction based on the qualitative measures they have not assessed the satisfaction of patients in terms of : availability of prescribed drugs , diagnostics availability, is the required procedure or treatment provided to patients, Out of pocket expenditure etc. Is the patients will again come to the same hospital, what about the hygiene and toilets uses etc.

Reviewer #2: Abstract

Please add a recommendation to the conclusion section.

Introduction

1. Please sue complete words of SDGs and then use its abbreviation.

2. Please use the first capital letter when you use an abbreviation. For example, you should write “Refugee Health Center (RHC)”. Use same format in whole your paper.

Methodology

1. Please write the study design used for this study. It is a cross sectional prospective quantitative study.

2. How the study settings were chosen? You have chosen them based on province or based on the RHTCs?

3. Please write the inclusion criteria clearly. What do you mean of “Participants who received services in RHCs”? What do you mean of services? What type of diseases were considered? How many time a patient should receive treatment or services to be included in this study? Only once or 2, 3…?

4. What were the exclusion criteria?

5. What was the sampling method used? What was the total population?

6. Please provide more information about data collection tool where you wrote “that was adopted from other humanitarian settings and pre-tested before implementation…”. What about its validity and reliability? Please add references.

7. How many section the questionnaire had? How many questions in each section? What were the questions version? Arabic or language...?

8. Please explain about study procedure. Who has collected data? Where the data was collected? In the RHTCs? How long took time to complete each questionnaire? Did you use information sheet and consent form before collecting data? In which languages? How about if a participant was not able to read or write?

9. Please use reference where you talk about scoring the satisfaction levels. What was the cut point to consider “Dissatisfied” or “Satisfied”.

Results

1. Please write 0.001 in table 2 and 3 instead of 0.000.

6. PLOS authors have the option to publish the peer review history of their article (what does this mean?). If published, this will include your full peer review and any attached files.

Reviewer #1: No

Reviewer #2: **Yes: **Masoud Mohammadnezhad

---

## [Author Response · Author response to Decision Letter 0]

30 May 2022

PLOS ONE's style requirements were followed in revising the manuscript 

2. Please include in your Methods section (or in Supplementary Information files) the participating hospitals/institutions. We note that you have reported significance probabilities of 0 in places. Since p=0 is not strictly possible, please correct this to a more appropriate limit, eg 'p<0.0001'.

Revised throughout the paper

3. Please provide additional details regarding participant consent. In the ethics statement in the Methods and online submission information, please ensure that you have specified 1) whether the ethics committee approved the verbal/oral consent procedure, 2) why written consent could not be obtained, and 3) how verbal/oral consent was recorded. If your study included minors, please state whether you obtained consent from parents or guardians in these cases. If the need for consent was waived by the ethics committee, please include this information.

The Ethics statement has been revised as below

Ethics statement 

The patient survey and its procedures including the participant consent process were reviewed and approved by the WHO Ethical Review Committee, Gazi University Ethical Board and the Ministry of Health Ethical Board in Turkey. The consent form was read to all participants that met the inclusion criteria and the response was recorded before administering the interview to only those that agreed to participate in the survey. Oral instead of written consent was sought because of the high illiteracy level in the sample population. 

The study involved minors, but only adult parents or guardians were interviewed according to the participant consent procedures approved by the ethical committees. 

4. Please include additional information regarding the survey or questionnaire used in the study and ensure that you have provided sufficient details that others could replicate the analyses. For instance, if you developed a questionnaire as part of this study and it is not under a copyright more restrictive than CC-BY, please include a copy, in both the original language and English, as Supporting Information.

More information about the survey has been added to the methodology

Furthermore, please provide additional information regarding the development and validation of the questionnaire.

Additional information about the questionnaire has been provided the methodology section

5. We note that the grant information you provided in the ‘Funding Information’ and ‘Financial Disclosure’ sections do not match. When you resubmit, please ensure that you provide the correct grant numbers for the awards you received for your study in the ‘Funding Information’ section.

6. We noted in your submission details that a portion of your manuscript may have been presented or published elsewhere. [Some of the results of this study are published in a WHO report on Patient satisfaction and experience at migrant health centres in Turkey. This article is a more concise publication of the factors that influence patient satisfaction.] Please clarify whether this [conference proceeding or publication] was peer-reviewed and formally published. If this work was previously peer-reviewed and published, in the cover letter please provide the reason that this work does not constitute dual publication and should be included in the current manuscript.

Some of the results of the study were published in a WHO publication, this article seeks to provide concise evidence on factors that influence patient satisfaction, particularly in facilities providing healthcare to refugees targeting a scholarly or research audience. While the study design was peer-reviewed the final results were not subjected to rigorous peer review. As such publishing, this work in a peer-reviewed journal will give scholars and researchers evidence that has been peer-reviewed that will catalyze further research and application. 

7. We note that you have indicated that data from this study are available upon request. PLOS only allows data to be available upon request if there are legal or ethical restrictions on sharing data publicly. For more information on unacceptable data access restrictions, please see http://journals.plos.org/plosone/s/data-availability#loc-unacceptable-data-access-restrictions.

The data collected from this study is co-owned by the Ministry of Health Turkey and WHO. The WHO ethics committee required that data collected from the study be stored in WHO Turkey country office and be only used for the purpose of this study. A request to access this data can be sent to the WHO representative in Turkey at eurowhotur@who.int

Reviewers' comments:

Reviewer's Responses to Questions 

Comments to the Author

1. Is the manuscript technically sound, and do the data support the conclusions?

Reviewer #1: Partly

Reviewer #2: Yes

2. Has the statistical analysis been performed appropriately and rigorously? 

Reviewer #1: Yes

Reviewer #2: Yes

3. Have the authors made all data underlying the findings in their manuscript fully available?

Reviewer #1: Yes

Reviewer #2: Yes

4. Is the manuscript presented in an intelligible fashion and written in standard English?

Reviewer #1: Yes

Reviewer #2: Yes

5. Review Comments to the Author

Reviewer #1: Healthcare of refugees is a major humanitarian concern and this type of studies should be encouraged to understand the delivery of healthcare services to refugees. Findings of the study resonates the same theoretical principles those required for the batter patient satisfaction.

there is a few concerns which needs further details -

1. Though sample size to measure the patient satisfaction is sufficient but still need more details on the sample size calculation as how they come to sample size of 4548, do they have calculated the sample size for each province separately. How they choose patients visiting health facilities (sampling technique), do they use design effect?

A proportional stratified sampling approach was followed to estimate the required sample size based on the total patient consultations in each province (strata) from 2017 until March 2019. A minimum sample size of 4460 individuals, was calculated using WinPepi (version 11.65) with a 95% CI, 0.05 error margin, and 20% loss to follow-up. The sample size was then distributed proportional to the volume of consultations in each of the 16 provinces and type of RHCs. The RHCs where data was collected were randomly selected, from a list of RHCs provided by the Ministry of Health. In each RHC, participants were systematically recruited in the reception areas on regular working days at an interval calculated from the average daily patient load of the facility. Of the 4665 people who met the recruitment criteria and were asked for an interview, 117 refused; therefore, 4548 participants were included in the study.

2. Study mostly captures the satisfaction based on the qualitative measures they have not assessed the satisfaction of patients in terms of availability of prescribed drugs, diagnostics availability, is the required procedure or treatment provided to patients, Out of pocket expenditure etc. Is the patients will again come to the same hospital, what about the hygiene and toilets uses etc.

The measures used in the study are based on a literature review and the context of health service provision for Syrian refugees in Turkey. For instance, in the context of RHC, drugs are not dispensed in RHC rather in community pharmacies that are linked to RHC, and all services are provided free of charge in RHCs. We generally assessed satisfaction with diagnostics and treatment in questions related to communication and the quality of services in regard to examination, explaining medical condition and medical tests and medicine prescription. 

Reviewer #2: Abstract

Please add a recommendation to the conclusion section.

The conclusion section was edited to include a recommendation

Introduction

1. Please sue complete words of SDGs and then use its abbreviation. Written in full

2. Please use the first capital letter when you use an abbreviation. For example, you should write “Refugee Health Center (RHC)”. Use same format in whole your paper. Harmonized across the manuscript

Methodology

1. Please write the study design used for this study. It is a cross sectional prospective quantitative study. Included

2. How the study settings were chosen? You have chosen them based on province or based on the RHTCs?

Provinces were selected for sampling the RHCs for the study. Sixteen provinces with the highest number of patient consultations were selected for better representation of patients receiving services from RHCs. In each province, RHCs by type were randomly selected from a list provided by the Ministry of Health.

3. Please write the inclusion criteria clearly. What do you mean of “Participants who received services in RHCs”? What do you mean of services? What type of diseases were considered? How many time a patient should receive treatment or services to be included in this study? Only once or 2, 3…?

Participant inclusion criteria and choice of study setting has been expounded in the methodology section

4. What were the exclusion criteria?

Patients under 18 years were excluded if they did not have an adult caregiver or guardian

5. What was the sampling method used? What was the total population?

6. Please provide more information about data collection tool where you wrote “that was adopted from other humanitarian settings and pre-tested before implementation…”. What about its validity and reliability? Please add references.

7. How many section the questionnaire had? How many questions in each section? What were the questions version? Arabic or language...?

8. Please explain about study procedure. Who has collected data? Where the data was collected? In the RHTCs? How long took time to complete each questionnaire? Did you use information sheet and consent form before collecting data? In which languages? How about if a participant was not able to read or write?

9. Please use reference where you talk about scoring the satisfaction levels. What was the cut point to consider “Dissatisfied” or “Satisfied”.

The methodology section has been expanded to include the above feedback

Results

1. Please write 0.001 in table 2 and 3 instead of 0.000.

Revised throughout the paper

---

## [Decision Letter · Decision Letter 1]

26 Aug 2022

Factors affecting patient satisfaction in Refugee health centers in Turkey

PONE-D-21-25728R1

Dear Dr. Zikusooka,

We’re pleased to inform you that your manuscript has been judged scientifically suitable for publication and will be formally accepted for publication once it meets all outstanding technical requirements.

Kind regards,

Alok Ranjan

Academic Editor

PLOS ONE

Additional Editor Comments (optional):

Reviewers' comments:

Reviewer's Responses to Questions

**Comments to the Author**

1. If the authors have adequately addressed your comments raised in a previous round of review and you feel that this manuscript is now acceptable for publication, you may indicate that here to bypass the “Comments to the Author” section, enter your conflict of interest statement in the “Confidential to Editor” section, and submit your "Accept" recommendation.

Reviewer #2: All comments have been addressed

2. Is the manuscript technically sound, and do the data support the conclusions?

Reviewer #2: Yes

3. Has the statistical analysis been performed appropriately and rigorously? 

Reviewer #2: Yes

4. Have the authors made all data underlying the findings in their manuscript fully available?

Reviewer #2: Yes

5. Is the manuscript presented in an intelligible fashion and written in standard English?

Reviewer #2: Yes

6. Review Comments to the Author

Reviewer #2: Thank you for addressing my comments. There is no new comments and this paper can be published. Good luck

7. PLOS authors have the option to publish the peer review history of their article (what does this mean?). If published, this will include your full peer review and any attached files.

Reviewer #2: **Yes: **Masoud Mohammadnezhad

---

## [Editor Report · Acceptance letter]

6 Sep 2022

PONE-D-21-25728R1 

Factors  affecting  patient satisfaction in  Refugee Health Centers in Turkey 

Dear Dr. Zikusooka:

I'm pleased to inform you that your manuscript has been deemed suitable for publication in PLOS ONE. Congratulations! Your manuscript is now with our production department. 

Kind regards, 

on behalf of

Dr. Alok Ranjan 

Academic Editor

PLOS ONE